# Re-Thinking Soil Bioengineering to Address Climate Change Challenges

**Slobodan B. Mickovski** 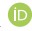

Built Environment Asset Management Centre, Glasgow Caledonian University, Glasgow G4 0BA, Scotland, UK; Slobodan.Mickovski@gcu.ac.uk

**Abstract:** Soil bioengineering includes the sustainable use of vegetation for civil engineering purposes, including addressing climate change challenges. Previous research in this area has been focused on determination of the strength and stability that vegetation provides for the soil it grows in. The industry, on the other hand, has concentrated on mainly empirical approaches in the design and construction of nature-based solutions. The aim of this paper is to attempt a reconciliation of the scientific and technical aspects of soil bioengineering with a view of proposing broad guidelines for management of soil bioengineering projects aimed at combatting climate change and achievement of the United Nations Sustainable Development Goals (UN SDGs). More than 20 case studies of civil engineering projects addressing climate change challenges, such as erosion, shallow landslides, and flooding, were critically reviewed against the different project stages and the UN SDGs. The gaps identified in the review are addressed from civil engineering and asset management perspectives, with a view of implementing the scientific and technical nexus in the future. Recommendations are formulated to help civil engineers embrace the multidisciplinary nature of soil bioengineering and effectively address climate change challenges in the future.

**Keywords:** soil bioengineering; nature-based solutions; climate change adaptation; landslides; erosion





## 1. Introduction

In 2015, the United Nations (UN) set out 17 global Sustainable Development Goals (SDGs) with 169 sub-targets, most of which are directly or indirectly connected to some form of infrastructure [1]. The built environment and construction industry employs around 15% of the UK workforce, which can be compared to 27% of all industrial employment in Europe. Therefore, it is essential to understand that sustainability for the infrastructure can only be achieved if the elements associated with it are supported and sustained (e.g., support the human resources in their well-being and safety, their employment security, skills development, etc.). Part of this support, estimated at £7 bn/year for the UK [2], is expected to be delivered through the ongoing digital transformation of the infrastructure industry, although these benefits are aimed at the people for whom the infrastructure is built. However, as our built environment and the wider economy become more information-based, associated "softer" infrastructure is under pressure to keep pace. In particular, new frameworks, organizational structures, and business models are and will be required to understand, plan, manage, and regulate our infrastructure and the related data infrastructure.

Arguably, one of the most important UN SDG focuses on climate action, in terms of strengthening the resilience and adaptive capacity of infrastructure to climate-related hazards and natural disasters in all countries. These hazards include shallow landslides, flooding, and erosion, which are expected to become more frequent in the near future. These hazards can originate outside of the infrastructure or within it, and affect not only infrastructure users, but also construction workers and the general public. Because of the increased likelihood and the potential severity of these hazards, there is a need to integrate

climate change measures into national and international policies, strategies, and planning, but also to improve education, awareness-raising, and human and institutional capacity on climate change mitigation, adaptation, and impact reduction. Integrated assessments of threats to natural and man-made infrastructures and implementing ecosystem-based climate action measures (e.g., [3,4]), such as operationalization of the green infrastructure and employment of nature-based solutions [5], are needed in order to address climate change challenges while, at the same time, minimizing the risks to infrastructure and the public.

Green infrastructure (GI) is usually defined as a network of natural and semi-natural areas that is strategically planned, designed, and managed, together with other environmental features, to deliver a wide range of ecosystem services [6,7]. This network, in general, comprises "green" (where terrestrial ecosystems are present, usually referring to the live vegetation forming the structure, elements, or parts of the structure) and/or "blue" (aquatic ecosystems) spaces, and other, sometimes "grey" (traditional engineering structures) features. GI can be found inland, in urban or rural settings, or close to aquatic surroundings (fluvial, coastal, etc.). The difference between GI and the traditional "grey" infrastructure is that the GI is rich with biodiversity, which, in turn, can be used to perform a variety of useful functions, including engineering, for the benefit of not only nature, but also people and the economy.

Assigning an engineering function to the vegetation within a "live" structure or "green" infrastructure system has been the basic concept of soil or soil bioengineering [8], sometimes also termed soil- and water bioengineering [9]. Using this concept, which combines scientific and practical knowledge and skills for ecosystem management and benefits to the natural and man-made environment, vegetation is employed as a building material, contributing to natural hazard control (e.g., erosion, flooding, landslides; [10]; Figure 1) and mitigation of the consequences of these hazards (e.g., ecological and engineering restoration of degraded lands, disturbed slopes, etc.). Within this concept, nature-based solutions (NBS) are employed for the sustainable management and use of nature for tackling socio-environmental challenges including climate change, water security, water pollution, food security, human health, and disaster risk management. However, despite its many years of application and multiple designations, soil bioengineering has not yet reached the maturity beyond which it can be generalized to the application domains in which there is already materialized experience (Figure 1), allowing for its extension to other contexts of sustainable civil engineering.

The current experience shows that ecosystem-based approaches, such as GI, nature-based solutions, soil- and water bioengineering measures, and disaster risk reduction measures, are cost-efficient policy tools, but they are not used to their full extent [7]. Their potential should be further strengthened at national and transnational levels, perhaps through integration of the engineering components (e.g., standards, best practice, etc.) in the existing policy instruments.

The needs of the soil bioengineering practitioners, the associated construction professionals, and the scientific/academic community have been reviewed and investigated in some depth in the last decade [9,11,12]. The discrepancy in the needs of these stakeholders and the current knowledge was shown to stem from the lack of knowledge transfer between academia and the industry, as well as the lack of reconciliation between sustainable ecological restoration and traditional, engineering hazard control measures applied on civil engineering infrastructure. These recent reviews did not consider the engineering design aspects in detail, and did not propose specific actions aimed at the industry, which would be needed to achieve a reconciliation of the needs. While comprehensive and relevant to the construction industry and research, these reviews pre-date the UNSDG and the life cycle analysis, which could help in resolving the potential differences between the stakeholders.

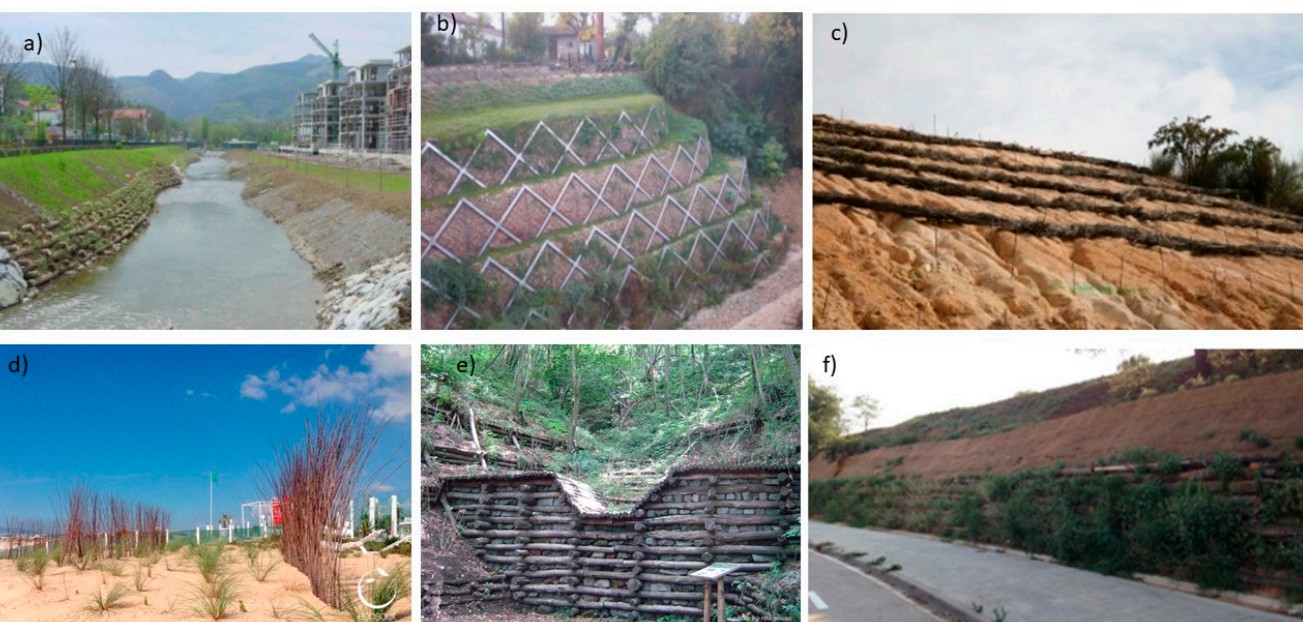

**Figure 1.** Typical soil bioengineering applications. (**a**) Live crib walls and hydroseeding for streambank protection in an urban area in Spain (Photo: Paola Sangalli, ecomedbio.eu); (**b**) live retaining wall and selected planting at the toe in Italy (Source: Jemmbuild Sarl; ecomedbio.eu); (**c**) live fascines for erosion protection in Spain (Source: Albert Sorolla, Naturalea.eu); (**d**) live erosion barriers for sand dune protection in Portugal (Source: Aldo Freitas, esocalix.pt); (**e**) vegetated check dams for debris flow protection (Source: Rita Sousa, ecosalix.pt); (**f**) live crib wall, hydroseeding, and biodegradable road embankment protection in Spain (Source: Albert Sorolla, Naturalea.eu).

The aim of this study is to offer a perspective on the engineering and scientific background of soil bioengineering based on the personal experience of the author. With this, this study also aims to offer a vision of soil bioengineering as a branch of civil engineering, which can not only provide effective solutions for combatting climate change, but also help in the achievement in UN SDGs. To achieve this, a set of successful and less successful climate adaptation case studies across Europe are mapped against the UN SDGs and analyzed against the project life cycle stages, highlighting the existing engineering and scientific knowledge as a basis that civil engineers can build on with innovation in order to respond to global climate change challenges.

## 2. Materials and Methods

To achieve the aim of this study, the research followed a pragmatic approach, drawing on published information and personal/professional experience to establish a representative set of soil bioengineering projects to be analyzed against existing research and best practice, policy, and regulations [13]. When developing the methodological approach for this study, care was taken not to enter into a comprehensive review of the literature and research on the subject, so as not to replicate similar valuable work [5,9–12].

A high-level case study analysis was then carried out on 22 case studies reported as part of the EU-funded ECOMED (www.ecomedbio.eu, accessed on 15 February 2021) project, which aimed at the promotion of soil and water bioengineering in the Mediterranean region. The case studies analyzed here are located in seven European countries (Figure 2) and represent projects where vegetation or NBS were used to provide soil retention, reinforcement, and stability in a fluvial, coastal, or inland slope environments, usually representative of green infrastructure. Details of the location and the type of case study analyzed are shown in Table 1. The case studies analyzed here sought to outline the types of problems associated with the application of soil bioengineering measures, and to establish an understanding of current practices in Europe. This dataset is further representative because of the consideration of both natural and man-made slopes, as well as participation

of a large range of stakeholders in the projects, including public companies, small and medium size enterprizes, local/national authorities, academia, research institutions, and the general public.

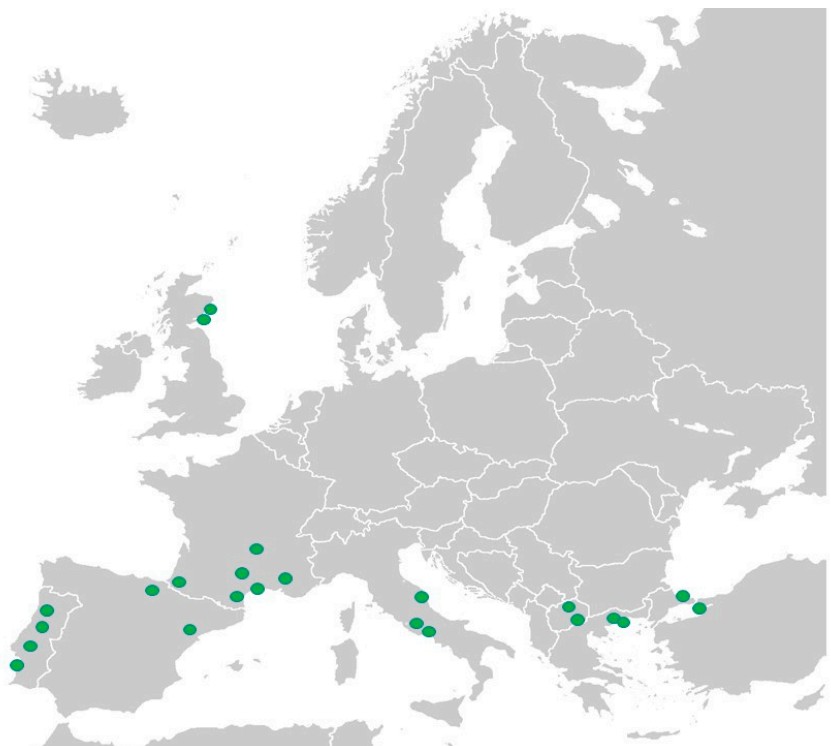

**Figure 2.** Geographic location of the analyzed case studies.

The case studies were thematically analyzed using the UN Sustainable Development Goals in order to not only highlight the benefits of the soil bioengineering approach, but also to expose the areas for future development and enhance the approach to be in reach of civil engineering research. To explore the details of each project covered with the case studies, a questionnaire survey and semi-structured interviews were carried out with at least one stakeholder (e.g., client, designer/engineer, planner, contractor, or user) per case study, with key themes emerging from the responses of the parties involved. The questionnaire survey was designed to map the project stages at which the soil bioengineering approach was applied in each case study. The interviews were designed to gauge the views of the interviewees on the current state-of-the-art soil bioengineering application, and the lessons that may be learned from the application in each case study. The responses from the questionnaires and interviews were thematically analyzed and triangulated in conjunction with the available project records and documentation for each case study. This approach helped to crystalize the need of soil bioengineering projects, ascertain the typical project workflow (project stages), and identify any gaps in the knowledge from the point of view of the stakeholder involved in such a project.

Applying the UN SDGs as a framework to each case study and critically analyzing the studies from the project lifetime point of view allowed for the identification of potential obstacles for the future application of soil bioengineering and lessons for developing soil bioengineering in the future. As this research was aimed at providing a vision for soil bioengineering in the future, this descriptive and explanatory approach provided a pragmatic analytic framework around which to explore the complexity and challenges of delivering soil bioengineering projects, while analyzing the design and stakeholder engagement against the decisions and policies during the lifetime of the project.

**Table 1.** Responses from the case study questionnaire survey and the interviews, mapped against the United Nations Sustainable Development Goals (UN SDGs). SDG1: no poverty; SDG2: zero hunger; SDG3: health and well-being; SDG4: quality education; SDG5: gender equality; SDG6: clean water and sanitation; SDG7: affordable clean energy; SDG8: work and economic growth; SDG9: innovation and infrastructure; SDG10: reduced inequality; SDG11: sustainable cities and communities; SDG12: responsible production and consumption; SDG13: climate action (resilience); SDG14: life below water; SDG15: life on land; SDG16: peace, justice; SDG17: institution partnership.

| Case Study Short Description | Environment | Location | Country | Feasibility | Design | Award | Mobilisation | Construction | Demobilisation | Monitoring | Long Term | UN SDGs Considered |
|---|---|---|---|---|---|---|---|---|---|---|---|---|
| National park | Slope | Gaeta | Italy | | | | | | | | low vegetation cover | 3,8,9,11,13,15,16,17 |
| Riverbank | Slope | Melfa | Italy | | | no maintenance contract | | design change during construction | | monitoring site visit every 6 months | | 3,9,11,13,15, 17 |
| River channel | Fluvial | Baztan | France | | climate change resilience based on hydrology report | relatively long period between design and construction | | well qualified workforce used | | | | 3,6,9,11,13,15,17 |
| Riverbank | Fluvial | Garonne | France | wide range of stakeholders involved | design based on ecological/ social/ technical report | | | lack of qualified workforce | | | | 3,6,9,11,13,15,17 |
| Riverbank | Fluvial | Longes Aygues | France | | no geotechnical report | | | partial failure of the constructed measure | | | | 3,6,9,11,13,15, 17 |
| Riverbank | Fluvial | Arize | France | | | no maintenance contract | | | | | incrwase in erosion due to trampling | 3,6,9,11,13,15,17 |
| Riverbank | Fluvial | Hers | France | | | no maintenance contract | | | | no monitoring contract | soil creep and erosion in long term | 3,6,9,11,13,15,17 |

**Table 1.** *Cont.*

| | | | | | | | | | |
|---|---|---|---|---|---|---|---|---|---|
| River channel | Fluvial | Artia | Spain | | | | | adjacent land and land use pose future risks | 3,6,9,11,13,14,15,17 |
| Roadside slope | Slope | Ripe | Italy | | | | low vegetation cover immediately after construction | local plant species affectd by climate change | 3,8,9,10,11,13,17 |
| Torrent catchment | Slope | Thasos | Greece | prescriptive design only; no field testing/measurements | no maintenance contract | | | no monitoring | adjacent land poses future risks; GIS should be used | 3,9,11,13,17 |
| Marble quarry restoration | Slope | Drama | Greece | | no maintenance contract | manual seeding/planting only resulting in low cover | | no monitoring | low plant density | 3,8,9,11,13,15,17 |
| Motorway cutting | Slope | Nogaevci | Macedonia | | limited funding | works not fully implemented | | no monitoring | | 3,8,9,10,11,13,17 |
| Motorway cutting | Slope | Gevgelija | Macedonia | | limited funding | works not fully implemented | | no monitoring | washout/erosion | 3,8,9,10,11,13,17 |
| Infrastructure cutting | Slope | Kartaltepe | Turkey | | | | | no/limited monitoring contract | | 3,8,9,10,11,13,17 |

Table 1. *Cont.*

| | | | | | | | | | | | |
|---|---|---|---|---|---|---|---|---|---|---|---|
| Riverbank | Fluvial | Couros | Portugal | | prescriptive design; no investigation | no maintenance contract | lack of qualified workforce | no quality control | no monitoring | slope failures and erosion after construction | 3,6,9,11,13,15,17 |
| Natural roadside slope | Slope | Albergaria | Portugal | | prescriptive design; no investigation | no maintenance contract | lack of qualified workforce | | no monitoring | | 3,8,9,10,11,13,15,17 |
| Riverbank | Fluvial | Argoncilhe | Portugal | | prescriptive design; no investigation | no maintenance contract | lack of qualified workforce | | no monitoring | slope failures and erosion after construction | 3,6,9,11,13,14,15,17 |
| Beach dunes | Coastal | Guincho | Portugal | | prescriptive design; no investigation | no maintenance contract | lack of qualified workforce | | no monitoring | | 3,9,11,13,14,15,17 |
| Coastal slope in a bay | Coastal | Stonehaven | Scotland | | | | lack of qualified workforce | | groundwater monitoring only | adjacent land and land use pose risks | 3,9,11,13,15, 17 |
| Coastal slope in a bay | Coastal | Catterline | Scotland | | | | | | specified inspection and limited monitoring | adjacent land and land use pose risks | 3,9,11,13,15, 17 |
| Riverbank | Fluvial | Tenes | Spain | | prescriptive design; no calculations; local standards used | | qualified personnel mobilised early | | | | 3,6,9,11,13,14,15, 17 |
| Beach dunes | Coastal | Terkos | Turkey | no engineering plan of works | | no maintenance contract | | | ocassional monitoring | | 3,9,11,13,17 |
| River estuary waterfront | Fluvial | Alverca | Portugal | | prescriptive design; no investigation | no maintenance contract | lack of qualified workforce | no quality control | no monitoring | slope failures and erosion afterc onstruction | 3,6,9,11,13,15,17 |

## 3. Results

The responses from the questionnaire survey and the interviews helped in mapping the case study projects against the UN SDGs (Table 1). The mapping showed that the full range of soil bioengineering problems analyzed contributed towards achieving SDG3, SDG9, SDG11, SDG13, SDG15, and SDG17. This was because of the nature of the soil bioengineering approach, which is usually applied as an innovative, non-traditional approach to minimize the risks to life and property, while employing live vegetation which, in turn, provides a sustainable and resilient habitat for a number of terrestrial or aquatic species. Most of the analyzed case study projects arose from the need to prevent or mitigate climate change effects (e.g., flooding, erosion, shallow landslides, and desertification) in urban or rural settings; although, this was not always specified in the respective project task briefs. In order to provide healthy, sustainable cities and communities, soil bioengineering was chosen as an innovative approach to help the infrastructure cope with climate change challenges. These projects were usually delivered as a cooperation between a number of institutions, individuals, and the general public, showing achievement of SDG17 through institution partnership. In all of the reviewed case studies, soil bioengineering professionals worked in partnership with different institutions (e.g., local authorities, academia, overseeing organizations, communities, and the general public) to employ "live" elements (vegetation) in the engineering solution, which provided long-term sustainability (e.g., enhanced biodiversity, was financially more viable, required less maintenance, provided more recreational space; was more aesthetically pleasing, etc.) for the solution as well as an engineering function (e.g., roots reinforced soil; vegetation lowered the pore water pressures in the soil; branches and crowns provided interception and attenuation of rainfall or wind erosion, etc.). The fluvial case studies contributed towards the achievement of SDG6 by employing the vegetation in filtration and/or sediment retention, thus improving the water quality. Specific case studies, such as the marble quarry restoration, the national park, or the beach dune stabilization, showed that the soil bioengineering approach can be used as a basis for providing a safe work or recreational space. Similarly, the road and motorway stabilization projects showed that this approach can effectively help the sustainability of the transport infrastructure which, in turn, can support the local or national economy by enhancing the interconnectivity within a region, and thus, reducing inequality (European Commission, 2019).

A summary of the feedback gained from the questionnaire survey and interviews is shown in Table 1. While every effort was made to record feedback for every project stage in each of the case studies analyzed, this proved impossible for various reasons. The number of "missing" UN SDGs in Table 1 is partially due to the type of contract the soil bioengineering project was awarded, the type of organization undertaking the soil bioengineering work (usually specialist sub-contractors, only partially involved in the project management), and the (non-) existence of records for specific project stages. The lack of responses on the feasibility stage question reflect the fact that soil bioengineering works are usually commissioned reactively, i.e., as a mitigation, rather than with a proactive idea of using the vegetation for engineering purposes. In the analyzed case studies, the designers often took the initiative to include wider information on the environment surrounding the project, but there were instances where basic geotechnical or hydrological studies were not undertaken, or the design did not include calculations because of the lack of knowledge or understanding of the effects of vegetation. The award stage of the analyzed case studies was characterized by using mostly standard forms of contract to fit a limited budget, which often means an omission of post-construction maintenance. Sparse information was available on the mobilization or demobilization stages, and most of the respondents identified the lack of a suitably qualified workforce as a potential obstacle to the success of the project. The most common issues identified during the construction stage of the analyzed projects were design changes connected to the available budget or quality control which, in turn, may have contributed to the limited success of some of the implemented designs. The majority of the respondents reported a lack of regular and

specific monitoring after the project completion which, in turn, negatively affected the performance of the implemented approach in the long term.

## 4. Discussion

The analysis of the soil bioengineering case studies, although limited in breadth, showed that the concept of soil bioengineering can be an innovative and viable approach towards combatting climate change effects associated with different types of infrastructure. Similar to the conclusions of the authors of [7], the results showed that a wider strategic approach for soil bioengineering is lacking, with only local implementation at a small scale, and without recognizing the potential economic and social benefits of using green, instead of grey, infrastructure solutions. However, although the analyzed projects were relatively small in scale, they can be used as examples of what can be done on a larger scale, which is one of the aims of the current European policies. In order for such projects to be eligible and prioritized for funding on a regional or transnational level, the soil bioengineering practitioners should focus on connecting the projects to the whole range of UN SDGs. This raises the awareness of the approach, which can increase the opportunities for soil bioengineering to be promoted within the civil/geotechnical engineering industry. To achieve this, in addition to combatting climate change effects, some focus should be potentially given to minimizing poverty and hunger, while sustainably producing/consuming cleaner energy (Figure 3).

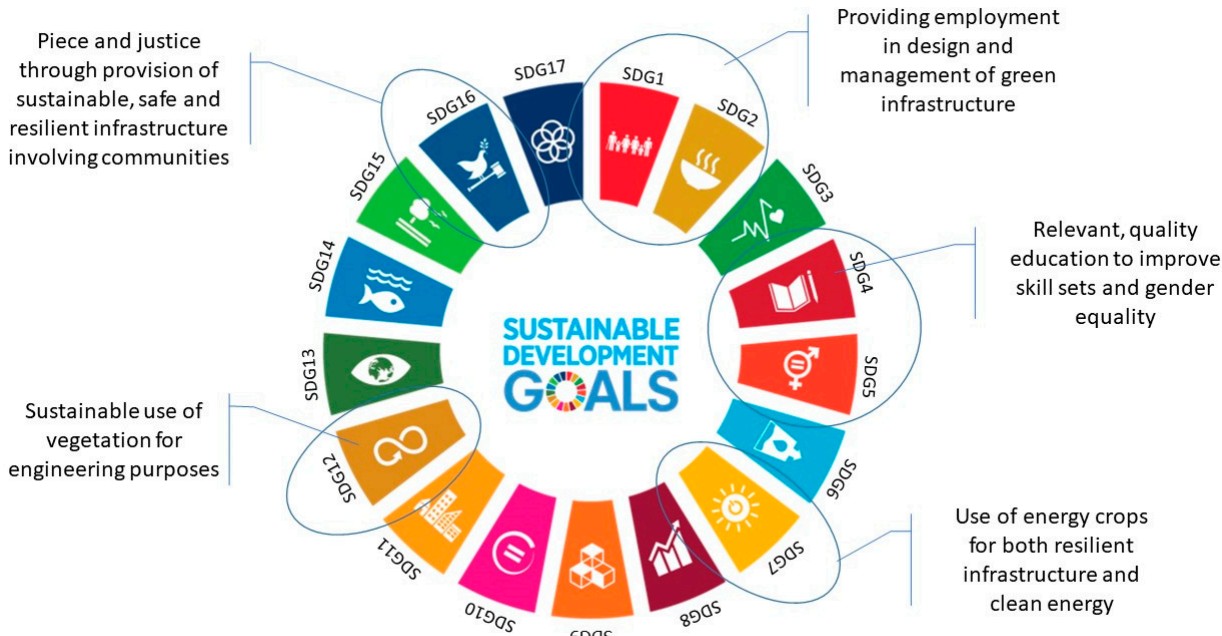

**Figure 3.** United Nations Sustainable Development Goals highlighting the needs and opportunities associated with soil bioengineering.

### 4.1. Sustainable Soil Bioengineering Project Management

The overwhelming evidence of the effects of soil bioengineering approach against the effects of climate change presented in these case studies and other literature [9,11], should motivate the engineers to embrace this approach from the inception/feasibility stage of the project (Figure 4). Once the engineering problem is identified and a connection is made with one or more climate change-related hazards, the range of potential engineering solutions should include at least one where an engineering function (e.g., drainage, soil reinforcement, sediment entrapment, wave attenuation, slope stability, etc.) is assigned to the "live" element. These considerations can be made on the basis of existing (un)successful case studies or experience, but must be contextualized with preliminary investigations spe-

cific to the location and the environment in which the soil bioengineering intervention is to be constructed. Such investigations would include not only the standard geo-hydrological and environmental investigations, but also investigations into the mechanism through which the "live" element provides the engineering function and the biological constraints associated with the potential solution. These can be easily appended to the standard set of preliminary investigations, and can allow not only for the easier quantification of the contribution of the "live" element, but also for knowledge transfer between academia and industry [11]. If the project is of a larger scale and/or conservation-based, a prioritized action framework [7] can be used as a tool for setting priorities for conservation and restoration at the regional or national level by including information on related, wider green infrastructure measures and by including engineering considerations that relate to the interaction between grey and green infrastructure. Within this strategic multiannual planning tool, aimed at determining the (co)-financing needs for measures that are needed to implement green infrastructure, links have to be made between the different EU funding programs and directives and the identified measures, aimed at the restoration and mainte-nance of natural habitats and species, whilst considering the economic, social, and cultural requirements at both a local and regional scale. In order to do this, the engineer needs to be able to map and assess the ecosystems where green infrastructure is planned/proposed to be built in accordance with the EU initiative on mapping and the assessment of ecosys-tems and their services (MAES). This is a good opportunity to enter soil bioengineering planning at a strategic level for civil engineers because, with the exception of Germany's "national GI concept", EU countries have not yet adopted national strategies specifically dedicated to GI. In terms of the contract award and mobilization for construction, it is important that the temporal scale of the vegetation effects is acknowledged throughout the planning for the full project life cycle. The focus should be the sustainability of the project in the long term, so adequate maintenance and monitoring carried out by suitably skilled professionals should be included. To assess the suitability of the professionals and the pro-posed long-term sustainability of the project, a planning tool in the form of a sustainability framework [13] can be used. The pre-qualification and quality submissions at the tender stage should comprise an assessment of the sustainability of the proposed actions by the tenderers against both generic and specific climate change challenges associated with the project, as well a commitment to minimum site disturbance during the demobilization and the defects correction period.

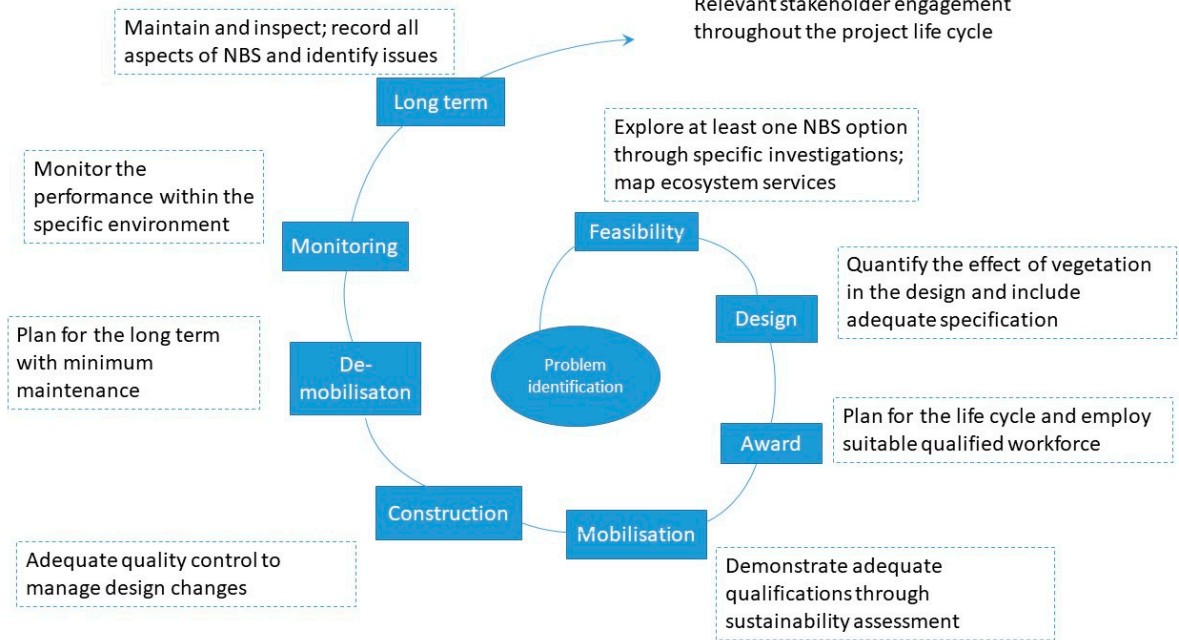

**Figure 4.** Needs and opportunities for soil bioengineering mapped against project life cycle stages.

The construction project stage, perhaps the most critical for the success of a soil bioengineering project, should be carried out by professionals suitably trained in the specifics of both civil engineering and plant science. As remarked by a number of the interviewees and with regard to the analyzed case studies, such a workforce would be able to respond to changes in the design or climate conditions rapidly, without compromising the overall sustainability of the project. Cognizance should be given to the vegetative period of the plants employed in the project, as well as to the potential interaction between the living environment (animals, insects, bacteria, etc.), the climate (wind, rain, period of insolation, etc.), and the inert and living construction materials [14]. The existing specifications for construction can be used ([10,15]) together with the "lessons learned" from open-air laboratories [5] in order to minimize the time needed for construction, but also to train the workforce.

### 4.2. Sustainable Soil Bioengineering Design

The research carried out on the effects of vegetation on the infrastructure response to climate change in recent years [16] should be used as part of the standard design procedure (Figure 4). The authors of [17] proposed detailed sampling and testing protocols, which can be used in a range of bio-geo-climatic regions in order to quantify the contribution of the vegetation to the engineering requirements of the solution. Similarly, specific detailed design protocols and routines [4,14,18] can be used in design for stability, while accounting for the spatio-temporal distribution and the hydrological and mechanical effects of the vegetation employed in the project. These protocols and routines are based on commonly accepted engineering principles, but are not specifically associated with any national or transnational standards, so future efforts should be focused on "translating: these into Eurocode language and synchronizing them with the relevant national standards (Bischetti et al., under review). Similarly, the existing protocols and routines can be used to develop an innovation in terms of a common soil bioengineering specification, which can enable transnational visibility and use.

### 4.3. Sustainable Soil Bioengineering Construction and Operation

Due to the specifics of design and construction with vegetation, special care must be taken during the early stages of operation and maintenance, when the vegetation needs to grow sufficiently to enable long-term engineering performance. Frequent inspections and surveys should be done in this stage in order to record the development of the green infrastructure; relevant data should be collected from these surveys and fed into numerical models that are able to assess the stability/performance of the "live" structure at different development stages [14]. All parties to the project should be aware that a longer defect correction period may be needed to accommodate the development period of the vegetation. This can be contractually regulated by including a maintenance (sub)-contract in the project documentation, which our research is lacking in current projects.

Associated with the maintenance of the soil bioengineering project, and potentially included in the maintenance contract, is the monitoring of the constructed measure, which should include integrated, real-time, multidisciplinary measurement and analysis of the parameters critical to the stability and sustainability of the installed measure. This would supplement the standard geotechnical or structural monitoring by incorporating relevant measurements of the vegetation parameters and would also include periodic sampling and in situ/laboratory testing (Figure 5). Draft protocols for the monitoring of NBS exist [17] and can be used together with topical reviews [5] in order to develop monitoring specifications for each specific project. The data collected during the maintenance/operation and monitoring stages should comply with the Building Information Modelling standards [19] and/or the Gemini Principles [20] in order to allow for a more efficient life cycle management of the installed measure and safe decommissioning at the end of life, both contributing to circular economy principles.

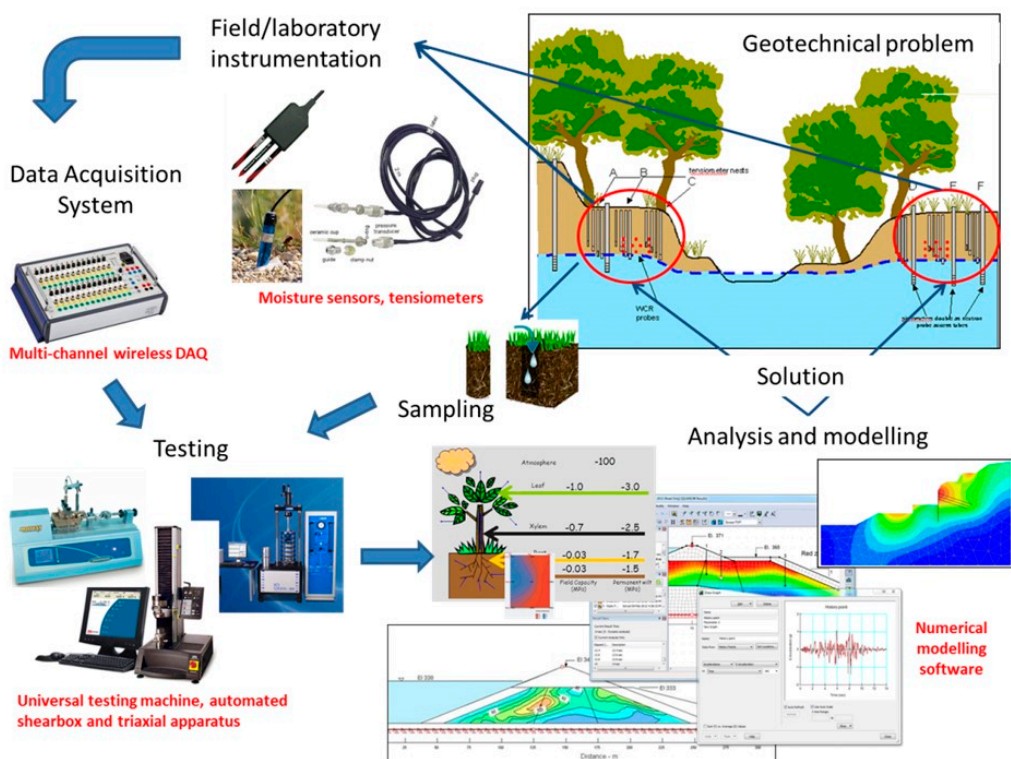

**Figure 5.** An example of integrated investigation, design, and monitoring using big data and smart concepts.

### 4.4. Soil Bioengineering Education

Finally, it is worth highlighting that the vision for the soil bioengineering sector outlined above can only become a reality if relevant and timely education is available. Undergraduate courses in soil bioengineering or soil and water bioengineering are sparse across Europe, although the application of the approach has been steady, especially in the Mediterranean and Alpine regions in the past two decades. This shows that more focus should be given to upskilling within the civil engineering and construction sectors to strengthen the knowledge base and promote innovation [7]. Masters courses and/or targeted and specific continuous professional development courses, where academia would provide the multidisciplinary theoretical (e.g., geotechnics/geology, hydrology/hydraulics, structural engineering, plant sciences, water resource management, ecology, data management, etc.), and the industry the practical (e.g., contracting, construction management, etc.), formation [20,21] may be the most practicable option. Similarly, considering green infrastructure as a socio-technical system, the communities and stakeholders affected by soil bioengineering works should also be educated in order to foster discussion between the infrastructure industry, government, and society about what outcomes are desired from infrastructure, and recognize those outcomes as the objectives for the industry.

### 4.5. Soil Bioengineering for Achievement of the Other UN SDGs

Exploring the bioengineering potential of energy crops that can be used for transport fuels or for heating/electricity generation is one potential innovative strategy. The fact that the soil strengthening properties of willows and poplars, often used as short rotation coppice, are relatively well-known [8] gives rise to the possibility of other vegetation (e.g., grass cut during the maintenance of the existing transport network; [22,23]) being used to provide resilience of the infrastructure against the effects of climate change, while also providing a source of sustainable energy or food. The maintenance and operation of the soil bioengineering assets should be associated with an increase in employment and wider stakeholder engagement which would, in turn, contribute towards the minimization of poverty and gender gaps, and, in cases where private residential properties managed

by the infrastructure owner (i.e., the case studies in France or Scotland, Figure 1) are at risk, may foster the sense of justice among the affected population. The success of a large-scale soil bioengineering project ultimately depends on the partnership of wide range of stakeholders, including the general public and the full supply chain, and the civil engineers in charge of such projects should have the awareness and knowledge of the specifics in order to successfully manage the project.

## 5. Conclusions

More than 20 soil bioengineering case studies were reviewed from the aspect of achieving UN SDGs across the project stages. The analysis showed that the soil bioengineering approach can provide innovative solutions for the infrastructure, which should resist the effects of climate change in terms of provisions of biodiversity on land and in water. One of the benefits of this approach was its use to prevent or mitigate climate change effects; although, this was not always specified in the respective project task briefs. Another benefit of this approach is the cooperation between a number of institutions, individuals, and the general public.

The selection of case study projects was robust in showing that the soil bioengineering approach fully contributed towards achieving six of the UN SDGs because of the nature of soil bioengineering approach. Although the case study project database was considered to be representative, it was limited to the personal experience of the author, and thus, it should be expanded in the future with the addition of more and more detailed case studies.

The drawback of the soil bioengineering approach may be that, due to the current scale of the projects, not all UN SDGs can be attempted. Future developments should see the application of forms of contract that are better suited for highlighting the achievement of multiple UN SDGs and employ a more advanced application of quality control/assessment throughout the project stages. Similarly, soil bioengineering should be used as a pro-active measure for minimizing climate change risks, and this can be achieved by more relevant and timely education as well as monitoring throughout the duration of the project.

In order to achieve these goals, soil bioengineering professionals need to work in partnership with academia and the communities affected by the works. Academia can provide the theoretical knowledge to promote soil bioengineering within the civil engineering and construction industries.

To successfully combat the growing climate change challenges and work towards sustainable and healthy communities, civil engineers should embrace the soil bioengineering approach in green infrastructure and nature-based solutions, while aiming at the achievement of multiple UN SDGs. To achieve these, however, civil engineers need to further enhance and implement the investigation, design, and construction procedures to include for the effects of vegetation. Similarly, more focus should be given to the maintenance and monitoring of the soil bioengineering structures in order to ensure a long and sustainable life cycle.

**Funding:** This research received no external funding.

**Institutional Review Board Statement:** Not applicable.

**Informed Consent Statement:** Not applicable.

**Data Availability Statement:** The data supporting reported results can be found at www.ecomedbio. eu (accessed on 15 February 2021).

**Acknowledgments:** The help and support of Alejandro Gonzalez-Ollauri and Guillermo Tardio and is greatly appreciated. The help from the ECOMED project partners and participants was invaluable in the compilation of the research data.

**Conflicts of Interest:** The author declares no conflict of interest.

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
