# Peer review of "Re-Thinking Soil Bioengineering to Address Climate Change Challenges"

_sustainability, doi:10.3390/su13063338_

Round 1

Reviewer 1 Report

In this paper more than 20 soil bioengineering case studies have been reviewed from the aspect of achieving of UN SDGs across the project stages. That analysis of the soil bioengineering case studies, showed that the concept of soil bioengineering can be an innovative and viable approach towards combatting the climate change effects associated with different types of infrastructure. It is very important in present time. In order to achieve these goals, soil bioengineering professionals need to work in partnership with academia and the communities affected by the works. 

The work is original, interesting, and good carried out. But in my opinion, the manuscript is not clearly written, but I don't feel qualified to judge about the English language and style.

The subject of the manuscript is consistent with the scope of the "Sustainability". I recommend publication of the work after minor corrections. The manuscript needs some improvements:

(1) Please give define for acronym "UN SDGs" in line 12.

(2) Please give more information about “Prioritised Action Framework” (line 250)

Author Response

Please find the Author's responses in blue below:

In this paper more than 20 soil bioengineering case studies have been reviewed from the aspect of achieving of UN SDGs across the project stages. That analysis of the soil bioengineering case studies, showed that the concept of soil bioengineering can be an innovative and viable approach towards combatting the climate change effects associated with different types of infrastructure. It is very important in present time. In order to achieve these goals, soil bioengineering professionals need to work in partnership with academia and the communities affected by the works.

I thank the Reviewer for this constructive comment. In the revised manuscript, I have stressed the need for collaborative work between the industry, academia, and communities, especially in the revised Conclusion section.

The work is original, interesting, and good carried out. But in my opinion, the manuscript is not clearly written, but I don't feel qualified to judge about the English language and style.

The manuscript was proofread by two native English academics which, I believe has improved the readability. I would like to stress that the manuscript is submitted as a Perspective/Commentary rather than the standard Research Article which, hopefully, justifies the difference in format.

The subject of the manuscript is consistent with the scope of the "Sustainability". I recommend publication of the work after minor corrections. The manuscript needs some improvements:

(1) Please give define for acronym "UN SDGs" in line 12.

The acronym has been described in full in the revised manuscript.

(2) Please give more information about “Prioritised Action Framework” (line 250)

European Commission (2019) defines the Prioritised action frameworks (PAFs) as strategic multiannual planning tools, aimed at determining the (co)-financing needs for measures that are needed to implement green infrastructure associated with different EU funding programmes and directives. The measures identified in a PAF should be aimed at restoration and maintenance of natural habitats and species whilst considering the economic, social and cultural requirements at a local and regional scale. This information is included in the revised manuscript together with a reference to the European Commission’s source document. 

Reviewer 2 Report

The topic is surely interesting and the manuscript is limguistically correct. The author should select the main class of the article (feature, communication). I think that a commentary is appropriate for this work. Literature review is incredibly limited. It should be expanded throughout the text. Figures and graphs are appropriate, but data sources and methodologies applied (even if simplified) should be described in a better form. The rationale of the study should be also clarified. Pros & cons should be expressed explicitly. Proposals for future studies are completely lacking.

Author Response

Please see the Author's response in blue below:

The topic is surely interesting and the manuscript is limguistically correct. The author should select the main class of the article (feature, communication). I think that a commentary is appropriate for this work.

I thank the Reviewer for the positive and constructive comment. I have submitted the manuscript as a ‘Perspective’ because it. Hopefully, provides a personal perspective based on a more of 20 years’ research in the field of ground bio-engineering. I am happy to change the category to ‘Commentary’ with the approval from the Editor.

Literature review is incredibly limited. It should be expanded throughout the text.

The literature review was deliberately kept close to the personal experience of the author in order to provide a broad empirical perspective and commentary of more than 20 years’ of research and practice in the field, rather than a general review of the published research on the topic. This has now been clarified in the Aim/objective of the revised manuscript and the personal perspective put in appropriate context with the inclusion of the various regulatory/normative literature. Having said this, I would be happy to consider adding to the reference list any relevant publications the Reviewer should volunteer to suggest.

Figures and graphs are appropriate, but data sources and methodologies applied (even if simplified) should be described in a better form.

I have added a new paragraph on the Materials/Methods (line 118-123) which, hopefully, provides both the rationale/justification and detail of the used methodologies (expert opinion, survey, interview, thematic analysis, mapping). Distinction is made between the approach adopted for this perspective/commentary and a more traditional review paper.

The rationale of the study should be also clarified.

The aim of the study has been reworded to read: ‘The aim of this study is offer a perspective on the engineering and scientific back-ground of soil bioengineering based on the personal experience of the author. With this, this study also aims to offer a vision of soil bioengineering as a branch of civil engineering which can not only provide effective solutions for combatting climate change, but also help in the achievement in UN SDGs’. The rationale stems from the critical overview of the past efforts (research and industry), detailed in lines 34-37, 48-52, 74-78, 91-94, 102-107.

Pros & cons should be expressed explicitly.

I thank the reviewer for this comment. The major pros and cons of the approach have now been explicitly stated in the Conclusions section. I believe that this revision resulted in ‘punchy’ take-home message from this paper.

Proposals for future studies are completely lacking.

Throughout the manuscript, rather than proposing future studies, I have recommended steps and direction for future development of this emerging subject from the perspective of an engineer/professional who would practice ground bio-engineering having been trained in a traditional discipline associated with ground bio-engineering (examples: lines 239-241, 243-246, 256-263,289-305, 312-314, 319-325, 330-341,  346-349, 361-363, 368-372, 386-390, 402-404, 406-411, 416-423). With this, I believe, I’ve kept to the expected format of a ‘perspective’ or ‘commentary’ paper rather than to the traditional research article which puts science/engineering (rather than the scientist/engineer) at the heart of the future developments. I have included the above argumentation in the revised text to avoid ambiguity and confusion.

Reviewer 3 Report

The topic of the manuscript “Re-thinking soil bioengineering to address climate change challenges” is interesting and author had a good idea for analysis of the UN Sustainable Development Goals.The critical review of more than 20 case studies is original, novel and important contribution to the knowledge of scientific and technical aspects of soil bioengineering addressing climate change challenges such as erosion, shallow landslides, and flooding. The subject is relevant, the methodologies are adequate, and the volume of reviewed data seems to be enough for publication. I have no hesitation in recommending publication following minor revision.

General comments:

Abstract: Abstract really presents summary, include key findings and the length of this part of the manuscript is appropriate.However, the author gives abbreviations in the abstract which are not explained. Should be extended and corrected.
Introduction: I consider that the structure of this section was well designed.Literature Review is adequate. Is effective, clear and well organized.
Material and methods: The methodology is well thought through.However, geographic location of the analysed case studiesshould be describedin more detail. The surveyed countries are admittedly given, but only in the results in Table 1.
Results and discussion: The results of the study are well presented. 
Conclusions: In my opinion, the conclusions are too general, should be more detailed. 
References: Should by extended, more postions. 

The aim, range and results were clearly defined and demonstrate a good scientific knowledge of the issues being discussed. The work contains appropriate analyses of the results. Presented review constitutes a source of important information about the soil bioengineering approach in green infrastructure and naturebased solutions.

Author Response

Please see the Author's response in blue below:

The topic of the manuscript “Re-thinking soil bioengineering to address climate change challenges” is interesting and author had a good idea for analysis of the UN Sustainable Development Goals. The critical review of more than 20 case studies is original, novel and important contribution to the knowledge of scientific and technical aspects of soil bioengineering addressing climate change challenges such as erosion, shallow landslides, and flooding. The subject is relevant, the methodologies are adequate, and the volume of reviewed data seems to be enough for publication. I have no hesitation in recommending publication following minor revision.

I thank the Reviewer for the favourable and supportive view.

General comments:

Abstract: Abstract really presents summary, include key findings and the length of this part of the manuscript is appropriate. However, the author gives abbreviations in the abstract which are not explained. Should be extended and corrected.

This has now been corrected in the revised manuscript.

Introduction: I consider that the structure of this section was well designed. Literature Review is adequate. Is effective, clear and well organized.

Minor changes to the aim and objective of the paper have been included in the revised manuscript.

Material and methods: The methodology is well thought through. However, geographic location of the analysed case studies should be described in more detail. The surveyed countries are admittedly given, but only in the results in Table 1.

I acknowledge this shortcoming and have now included a reference to Table 1 where the reader can find the necessary detail on the location and type of case study/problem.

Results and discussion: The results of the study are well presented.

I thank the Reviewer for this comment.

Conclusions: In my opinion, the conclusions are too general, should be more detailed.

I acknowledge this comment from the Reviewer. The conclusions have been revised to provide more focus.

References: Should by extended, more postions.

The references were deliberately kept close to the personal experience of the author in order to provide a broad empirical perspective and commentary of more than 20 years’ of research and practice in the field, rather than a general review of the published research on the topic. This has now been clarified in the Aim/objective of the revised manuscript and the personal perspective put in appropriate context with the inclusion of the various regulatory/normative literature. Having said this, I would be happy to consider adding to the reference list any relevant publications the Reviewer should volunteer to suggest.

The aim, range and results were clearly defined and demonstrate a good scientific knowledge of the issues being discussed. The work contains appropriate analyses of the results. Presented review constitutes a source of important information about the soil bioengineering approach in green infrastructure and naturebased solutions.

I thank the Reviewer for this positive and supportive comment.

Round 2

Reviewer 2 Report

Good revisions overall. Publishable manuscript. I suggest to classify the paper as a 'perspective'